# Excipient Impact on Fenofibrate Equilibrium Solubility in Fasted and Fed Simulated Intestinal Fluids Assessed Using a Design of Experiment Protocol

**DOI:** 10.3390/pharmaceutics15102484

**Published:** 2023-10-17

**Authors:** Bayan E. Ainousah, Ibrahim Khadra, Gavin W. Halbert

**Affiliations:** 1Department of Pharmaceutical Chemistry, Faculty of Pharmacy, Umm Al-Qura University, Makkah 21955, Saudi Arabia; baaunosah@uqu.edu.sa; 2Strathclyde Institute of Pharmacy and Biomedical Sciences, University of Strathclyde, 161 Cathedral Street, Glasgow G4 0RE, UK; ibrahim.khadra@strath.ac.uk

**Keywords:** fenofibrate, mannitol, polyvinylpyrrolidone, hydroxypropylmethylcellulose, chitosan, simulated fasted intestinal media, simulated fed intestinal media, equilibrium solubility, design of experiment

## Abstract

Solubility is a critical parameter controlling drug absorption after oral administration. For poorly soluble drugs, solubility is influenced by the complex composition of intestinal media and the influence of dosage form excipients, which can cause bioavailability and bioequivalence issues. This study has applied a small scale design of experiment (DoE) equilibrium solubility approach in order to investigate the impact of excipients on fenofibrate solubility in simulated fasted and fed intestinal media. Seven media parameters (bile salt (BS), phospholipid (PL), fatty acid, monoglyceride, cholesterol, pH and BS/PL ratio) were assessed in the DoE and in excipient-free media, and only pH and sodium oleate in the fasted state had a significant impact on fenofibrate solubility. The impact of excipients were studied at two concentrations, and for polyvinylpyrrolidone (PVP, K12 and K29/32) and hydroxypropylmethylcellulose (HPMC, E3 and E50), two grades were studied. Mannitol had no solubility impact in any of the DoE media. PVP significantly increased solubility in a media-, grade- and concentration-dependent manner, with the biggest change in fasted media. HPMC and chitosan significantly reduced solubility in both fasted and fed states in a media-, grade- and concentration-dependent manner. The results indicate that the impact of excipients on fenofibrate solubility is a complex interplay of media composition in combination with their physicochemical properties and concentration. The results indicate that in vitro solubility studies combining the drug of interest, proposed excipients along with suitable simulated intestinal media recipes will provide interesting information with the potential to guide formulation development.

## 1. Introduction

When developing oral formulations for poorly water-soluble drugs biopharmaceutics classification system (BCS) (classes II and IV) [1], it is important to consider solubility and dissolution as potential absorption extent and rate-limiting factors [2]. The in vivo bioavailability of poorly soluble drugs is chiefly solubility and/or dissolution limited and assessed using the drug’s dose-to-solubility ratio based on solubility in complex gastrointestinal milieu [3]. A drug’s equilibrium solubility in any fluid will be constant but changes between products, excipients and/or manufacturing processes, can influence solubility and dissolution leading to bioequivalence problems [4]. Approximately forty percent of the molecules being developed currently are practically insoluble in water [5], and consequently, researchers have turned their attention to developing in vitro methods of predicting drug solubility and dissolution rates in the gastrointestinal tract (GIT). According to the BCS, increasing the solubility of class II (low solubility, high permeability) compounds is likely to improve bioavailability [6]. Drug solubility can be influenced by multiple factors present in GIT media, such as bile salt and ingested food composition, since this leads to an increase in the concentration of bile salt and lecithin with an increase in drug solubility [7]. In addition, several approaches have been applied to modify formulations either chemically, by the adjustment of the pH through the use of a buffer or salt formation, or physically, by particle size reduction, the modification of the crystal habit or the formation of hydrophilic matrices (powdered mixtures of the drug and excipient). All these aforementioned approaches are applied with the aim of improving drug solubility and dissolution rates [6]. These latter approaches are likely to produce formulations that will experience increased bioequivalence issues [8].

Published studies have sought to advance the understanding of drug solubility in the GIT, using either sampled human intestinal fluids [9] or simulated intestinal media, prepared according to literature data on the composition of GIT fluids [10]. There is no overall coherence in fasted or fed simulated media recipes, component concentrations or agreement on which is the optimal media [10]. To assess the effects of the various gastrointestinal fluid factors and their interactions on the intestinal solubility of BCS class II drugs, a statistical design of experiment (DoE) approach has been applied exploring the composition of fasted or fed state simulated gastrointestinal media [11,12]. The two studies confirmed the feasibility of the DoE approach, providing solubility data in agreement with solubility values found in the literature [9] and quantifying differences in solubility between the fasted and fed states. In addition, the approach was able to determine the key media components or factors controlling drug solubility in simulated gastrointestinal media. However, as both DoE studies involved numerous experiments (66 in the fasted and 92 in the fed state), the experimental load was decreased by conducting a reduced experiment dual-range DoE study covering fasted and fed states in a single protocol [13]. This dual-range design with a reduced number of experiments (10 fasted + 10 fed) demonstrated the ability of the design to provide equilibrium solubility data in both fasted and fed states that are equivalent to those of previous studies and to identify the major factors influencing solubility. Minor factors and factor interactions could not be identified due to the reduced statistical resolution arising from the lower number of experiments [14,15].

The experiments reported in the previous paragraph were performed using solid drug in media to determine an equilibrium solubility value, based on the principle that equilibrium solubility is a fundamental value that can be applied throughout the oral absorption space [16]. However, in an oral dose product the drug is formulated with a range of additional excipients and then processed through multiple stages to provide the final dosage form. For poorly soluble drugs, attaining the spreading of the drug particles at the intestinal level is one of the most important formulation approaches applied to improve dissolution and solubility [17]. This is achieved by incorporating different types of excipients or carriers (water soluble, amphiphilic, or lipid soluble) in the final dosage form with the poorly soluble drug [18,19]. Therefore, exploring the solubility behaviour of poorly soluble drugs in the presence of excipients in gastrointestinal media would be a useful tool to assess the potential impact of formulation components on drug solubility and product performance. 

This study will assess the solubility impact of four excipients (see Appendix A for chemical structures) typically utilised in oral dosage formulations [20]. Mannitol, a non-reducing sugar, is widely used in pharmaceutical formulations as a diluent and in different production processes, such as wet granulation or direct compression. Hydroxy-propyl methylcellulose (HPMC, Hypromellose), a complex polysaccharide, is widely applied in oral formulations as a binder and matrix for controlled/extended release formulations. This semi-synthetic polymeric excipient is available in various grades; in this protocol, two grades will be assessed: Methocel E3 and E50 Premium LV™. Polyvinylpyrrolidone (PVP, Povidone), a synthetic polymer, is primarily used in solid dosage forms as a binder during wet granulation and also as a coating agent. This is also available in different grades or molecular weights, of which two grades K12 and K29/32 will be examined. Chitosan is a natural polysaccharide excipient that can vary in composition and has been applied in a range of oral pharmaceutical formulations types, such as mucoadhesive and colonic delivery systems. This is also a semi-synthetic polymer, but only a single grade will be examined.

This paper reports the modification of the dual-range DoE study [13] using refined component concentration levels to reflect the literature values and a single-protocol fasted and fed combination study. The aim is to investigate the effects of excipients on the equilibrium solubility of fenofibrate, a typical poorly soluble BCS class II drug. Fenofibrate was chosen as a model drug based on previous studies [11,12,13] since it is non-ionic and therefore not directly influenced by media pH. In previous studies, it has produced a 100-fold solubility variation and therefore is susceptible to changes in media composition. The range of excipients chosen and dual concentration levels studied aims to provide a broad spectrum of potential scenarios with a caveat that the systems must be considered indicative and not be linked to specific formulations or products. 

## 2. Materials and Methods

### 2.1. Materials

Sodium taurocholate (>97%), monosodium dihydrogen phosphate (100%), ammonium formate (>99.995%), formic acid (98–100%), sodium chloride (NaCl), potassium hydroxide (KOH, >85%), hydrochloric acid solution (HCl, analytical grade), acetic acid (>99.7%), cholesterol (>99%), chloroform (99.5%), fenofibrate, and chitosan (from crab shells, practical grade) were purchased from Sigma-Aldrich, Poole, Dorset, UK. Lecithin S PC (phosphatidylcholine from soybean, 98%) was supplied by Lipoid, Ludwigshafen, Germany. Sodium oleate (technical grade) was obtained from BDH Chemicals Ltd., Poole, England. Monoglyceride (glyceryl mono-oleate, >92% monoester, and 88% oleic acid) was supplied by CRODA. Polyvinylpyrrolidone (PVP, Plasdone TM K-12 and Plasdone TM K-29/32) were from Ashland, Singapore. Hydroxypropyl methylcellulose (HMPC, Hypromellose) (Methocel E3 and E50 premium LV) were obtained from Colorcon Ltd., Kent, UK. Mannitol (pharmaceutical grade) was obtained from Blackburn Distributions, Nelson, UK. All water used was ultrapure Milli-Q. Methanol and acetonitrile was purchased from VWR Prolabo Chemicals, Poole, UK. 

### 2.2. Dual Level Design of Experiment (DoE) and Data Analysis

For each media component (pH, sodium oleate, bile salt, lecithin, monoglyceride, cholesterol and BS/PL ratio), lower and upper limit concentration values for fasted and fed states were defined (Table 1). Using Minitab^®^ 17.2.1 and a custom experimental design, a 1/8 factorial DoE with seven factors (pH, sodium oleate, bile salt (BS), lecithin (PL), monoglyceride (MG), cholesterol and BS/PL ratio) and two levels (lower and upper limits, see Table 1) was constructed (16 experiments around the upper and lower levels plus two centre points) separately for the fasted and the fed states. Each DoE experiment was measured once to limit the total number of solubility experiments required. This approach has been adopted in the conduct of previous DoE studies [13,21].

When designing and analysing the DoE, only the main effects have been considered, and higher interactions of factors were not included. For each DoE, the magnitude of each individual factor’s effect on equilibrium solubility was determined by the standardised effect value. This value was used to determine whether these factors raised or lowered drug solubility. Due to the design and the low number of experiments, the standardised effect values calculated for individual factor effects in the fasted and fed state arms indicate a significant increase in drug solubility when it is greater than +2 and a decrease when it is less than −2. The Kolmogorov normality test was used in Minitab^®^ to assess the normality distribution of each data set, and the Friedman Test and Kruskal–Wallis test (Prism 9.5.1 on Mac OS 10.13.6) used to evaluate the non-parametric difference between data sets. 

### 2.3. Equilibrium Solubility Measurement

The media preparation and equilibrium solubility measurement method has been applied in previous DoE studies [11,12,13].

The concentration of each stock solution has been designed to be 15 times greater than the upper limit concentration value required for the DoE, with the exception of oleate, for which only a 5 times concentration was possible (see Appendix A). 

#### 2.3.1. Preparation of Stock Systems

Preparation of Lipid Suspension

Sodium taurocholate, monoglyceride, lecithin and cholesterol were weighed and transferred into a flask, then 2 mL of chloroform was added to dissolve all the solid material. A stream of nitrogen gas was applied in order to remove the chloroform and to ensure the formation of a dried film. Water was added to reconstitute the dried film, and the solution was mixed to obtain a homogenous suspension, transferred to a 5 mL volumetric flask, and brought to volume with water. 

Preparation of Sodium Oleate Solution

Sodium oleate (1.90 g) was weighed and placed into a 50 mL volumetric flask, dissolved in water with the assistance of gentle heating (37 °C) to aid dissolution, and then made up to volume with water and kept under heat to aid solubilisation.

Preparation Buffer Solution

A concentration of 0.3 M monosodium dihydrogen phosphate buffer was prepared by adding 20.4 g into a 500 mL volumetric flask and making up to volume with water. This was split into two, and the pH was adjusted to 5 and 7 using aqueous 0.5 M HCl or 0.5 M KOH. 

Preparation of excipient stock solutions

Excipient solution: The appropriate amount of each excipient was weighed out (see Appendix A), transferred to a 25 mL volumetric flask, and brought to volume with water, with the exception of chitosan, which was dissolved in 0.1 M acetic acid and mixed overnight with a magnetic stirrer under heat.

#### 2.3.2. Preparation of Experimental Measurement Solutions

Preparation of Individual Design of Experiment Solutions

Individual experimental solutions were prepared following previous published protocols that have been demonstrated to successfully permit the determination of equilibrium solubility [11,12,13]. The solution was prepared by the addition of an excess amount (above the estimated solubility) of fenofibrate powder [13] to a centrifuge tube (15 mL Corning^®^ Centristar™ cap, polypropylene RNase/DNase free, non-pyrogenic) followed by the addition of each component of the simulated intestinal fluid media according to the run order generated by the DoE together with the excipient to be examined (see Appendix A). After all of the media components were added, the pH was adjusted to 5, 6 or 7 according to the run order using 0.1 M HCl or 0.1 M KOH, and tubes were capped and placed on an orbital shaker (OS 5 basic Yellowline, IKA, Staufen, Germany) for 1 h, after which the pH was readjusted if required. The tubes were then shaken in a tube rotator for 24 h at 40 rpm at 37 °C. After 24 h, a 1 mL amount was taken from each tube, transferred to a 1.5 mL Eppendorf^®^ tube then centrifuged at 15,000 rpm for 5 min. Following centrifugation, 0.5 mL of the supernatant solution was transferred to an HPLC vial to analyse drug solubility using HPLC, as will be discussed below. 

### 2.4. HPLC Analysis

Agilent Technologies 1260 Series liquid chromatography system with Clarity chromatography software was used. Gradient method: mobile phase A ammonium formate 10 mM, pH 3 in water; mobile phase B ammonium formate 10 mM, pH 3 in acetonitrile/water 9:1, time 0, 70%A:30%B, 3 min 0%A:100%B, 4 min 0%A:100%B, 4.5 min 70%A:30%B, total run 8 min. Column × Bridge C18 column/186003108/50 mm × 2.1 mm id. 5 μm was used. Fenofibrate (retention time 3.6 min) calibration curve (*n* = 6 concentrations) was run for the excipient-free DoE and each excipient concentration, with the lowest linear regression coefficient of 0.9979 for the low chitosan excipient concentration system. The method has been reported previously [13,21].

## 3. Results

### 3.1. Control Solubility Measurement

#### 3.1.1. Solubility Distribution

The results of the dual level equilibrium solubility measurements for the control experiment (no excipient) are presented in Figure 1 in comparison to published DoE results for fenofibrate in the fasted [11] and fed states [12]. 

**Figure 1 pharmaceutics-15-02484-f001:**
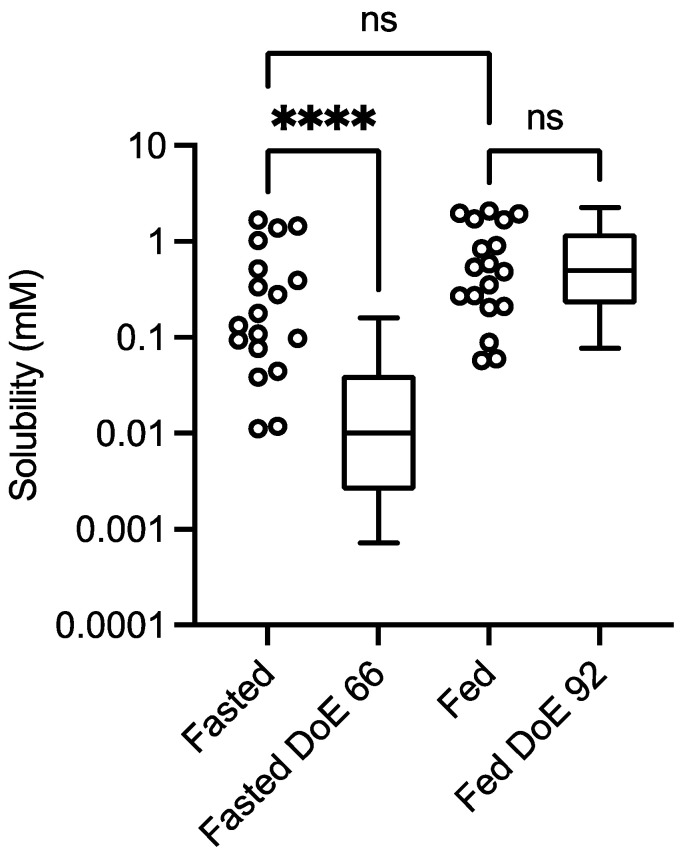
Excipient-free (control) solubility comparison. Design of experiment equilibrium solubility measurements. Values from this study: box and whisker plots from top to bottom, maximum value, 75th percentile, median, 25th percentile and minimum value. Fasted DoE 66 results from [11], fed DoE 92 from [12]. ns = No statistically significant difference; **** statistically significant difference, *p* < 0.0001. Kolomogorov normality test (KS) on the data sets: *p* < 0.05 indicates a non-normal distribution; fasted study arm, *p* < 0.0032, KS = 0.254; fasted DoE66 *p* < 0.0001, KS = 0.226; fed study arm, *p* < 0.02, KS = 0.2198; fed DoE92, *p* < 0.0001, KS = 0.1899.

#### 3.1.2. Solubility Influence of DoE Factors

The DoE measurements were analysed using Minitab to calculate an individual factor’s standardised effect on measured equilibrium solubility. This provides a value for the magnitude and direction of the factor’s effect and allows for a comparison between different factors (Figure 2). In this DoE protocol, only two factors were significant in the fasted state: pH, which had a negative impact on solubility, and sodium oleate, which had a positive impact on solubility.

**Figure 2 pharmaceutics-15-02484-f002:**
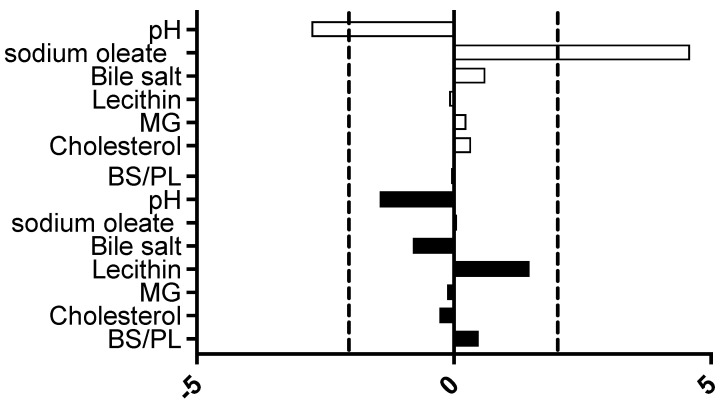
Excipient-free standardised effect analysis. Standardised effect values for individual DoE media factors on equilibrium solubility. DoE standardised effect values (*x*-axis) for individual factors (as listed in *y*-axis); 
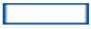
 represents fasted study arm; 
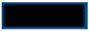
 represents fed study arm. Bar to the right of 0 on *x* axis is positive effect on solubility, bar length indicates the magnitude of the effect, vertical dashed black lines indicate statistical significance threshold (*p* < 0.05).

#### 3.1.3. Statistical Considerations

The impact of the various excipients and grades on fenofibrate equilibrium solubility is presented in Figure 3, Figure 4, Figure 5, Figure 6, Figure 7 and Figure 8. Each figure consists of a fasted and fed set of plots, arranged from left to right to present; the individual solubility values, as a bar chart, measured in each DoE tube in the absence or presence of the low then high excipient concentration; followed by two statistical comparisons of the three groups (control, and low and high excipient levels) of data. The first comparison is a non-parametric (see Section 4) matched Friedman comparison between the groups analysing all tubes as matched or repeated measures followed by a non-parametric Kruskal–Wallis comparison of each group with no matching. The Friedman and Kruskal–Wallis analysis were adopted for the following reasons: Due to the overall size of the study (108 samples peer excipient), each DoE media was measured once, and no statistical information on the solubility variability within a media is available. Therefore, a matched non-parametric Friedman comparison, which ranks individual solubility values, may be subject to random analytical variation, especially if there is a minimal solubility difference between tubes. To mitigate this issue a non-matched Kruskal–Wallis comparison of each group has also been added. Additional visualisation using the calculated solubility difference (Excipient Solubility ÷ Control Solubility) as a “heat map” (Figure 9 and Figure 10) of the solubility changes is also provided, highlighting which excipients and media compositions shift with respect to the control.

### 3.2. Impact of Excipients on Equilibrium Solubility

#### 3.2.1. Excipient Concentrations

Two excipient concentration levels in the final equilibrium solubility media were investigated, with the lower concentration representing systems in which the drug and formulation have been diluted within the tract contents and the higher concentration representing those in which the disintegration and or dissolution is in the initial stages. The concentrations were chosen to be approximately 10× (low) or 100× (high) the mean molar fenofibrate equilibrium solubility (0.8 mM) measured in previous DoE study [13]. In the case of HPMC E50 and chitosan, lower concentrations were used due to practical solubility and viscosity limitations of these excipients.

#### 3.2.2. Mannitol

The results for mannitol (Figure 3) show no significant solubility impact in the fasted state. The Friedman analysis detected a significant effect on solubility in the fed state between the control and high excipient concentration, but this was not detected in the Kruskal–Wallis comparison. Visually, with the exception of media number 13 in the fed state, there is minimal difference between the solubility measures in either state in the presence of low or high excipient concentrations. The minimal solubility change can be further visualised in the solubility difference heat maps of the fasted (Figure 9) and fed (Figure 10) states, in which mannitol has minimal response (with the exception of media 13 in the fed state) in comparison to the other excipients. In addition, mannitol does not change the significant factors impacting solubility (see Table 2 in comparison to Figure 2). 

**Figure 3 pharmaceutics-15-02484-f003:**
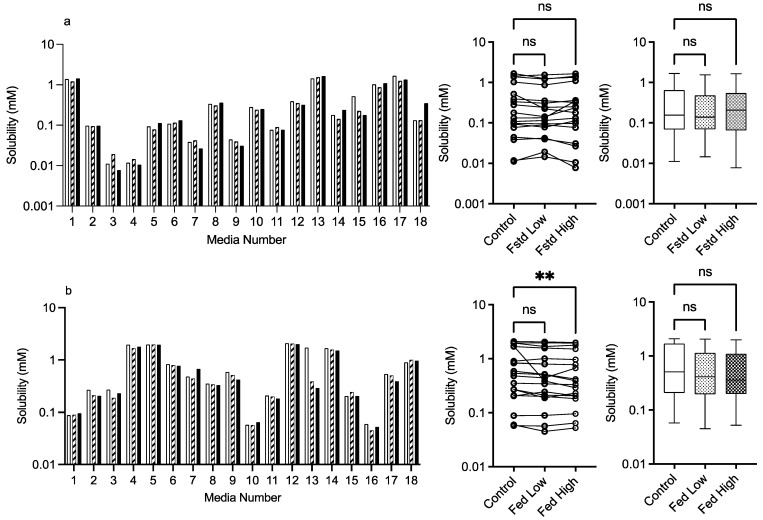
Mannitol. Top plots are for fasted media, bottom plots are for fed media. Plots from left to right: Bar graph shows individual media fenofibrate solubility measurements, 
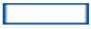
 control (no excipient), 
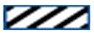
 low excipient concentration, 
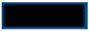
 high excipient concentration. Matched Friedman non-parametric comparison of groups (control, low excipient and high excipient concentration). Non-matched Kruskal–Wallis comparison of groups, 
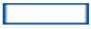
 control no excipient, 
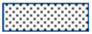
 low excipient concentration, 
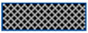
 high excipient concentration. Box and whisker plots—see legend in Figure 1. (**a**) ns = no significant difference, *p* > 0.05. (**b**). ** *p* = 0.0124.

#### 3.2.3. PVP K12 and K29/32

The results for PVP K12 and K29/32 are presented in Figure 4 and Figure 5, and visually there is a solubility increase due to the presence of the excipient in some, but not all of, the media systems. Media 18, for example, in the fasted state exhibits an obvious concentration-dependent effect of PVP K12 and K29/32 on fenofibrate solubility. The changes are statistically significant in the Friedman analysis of both the fasted and fed states but not in the non-matched Kruskal–Wallis test. In the Kruskal–Wallis presentation for both K12 and K29/32 in both states, it is visibly obvious that as the median solubility increases, the fasted box (75th to 25th percentile) shifts higher, but in both states, the overall range does not change. Figure 9 highlights the grade and excipient concentration-dependent increases in fenofibrate solubility in the fasted state, with Figure 10 showing that the solubility increase in the fed state is not as pronounced. In the fasted state the maximum increase is registered in the centre point media, while in the fed state, only media 4 with the K29/32 high excipient concentration provides a signal. For both PVP grades, there is no change in the standardised effect profile, as shown in Table 2. 

**Figure 4 pharmaceutics-15-02484-f004:**
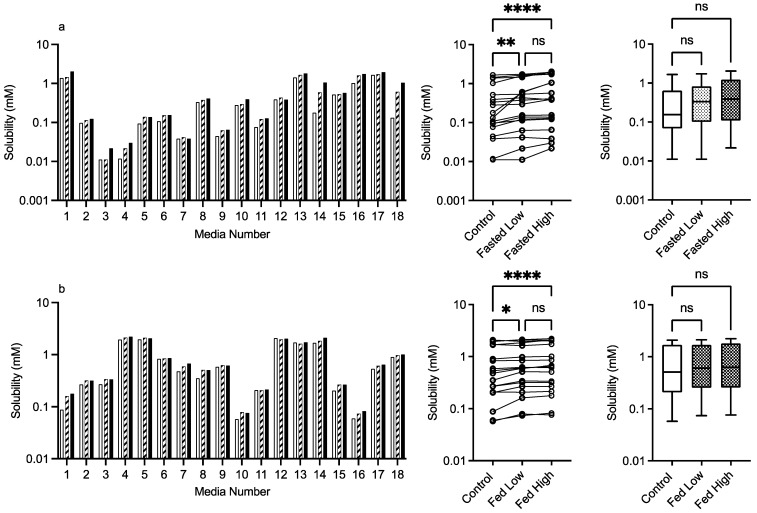
PVP K12. Top plots are for fasted media, bottom plots are for fed media. Plots from left to right: Bar graph shows individual media fenofibrate solubility measurements, 
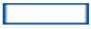
 control (no excipient), 
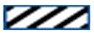
 low excipient concentration, 
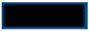
 high excipient concentration. Matched Friedman non-parametric comparison of groups (control, low excipient and high excipient concentration). Non-matched Kruskal–Wallis comparison of groups, 
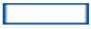
 control no excipient, 
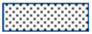
 low excipient concentration, 
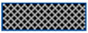
 high excipient concentration. Box and whisker plots—see legend in Figure 1 (**a**) ns = no significant difference, *p* > 0.05; ** *p* = 0.0081; **** *p* < 0.0001. (**b**) ns = no significant difference, *p* > 0.05; * *p* = 0.0138; **** *p* < 0.0001.

**Figure 5 pharmaceutics-15-02484-f005:**
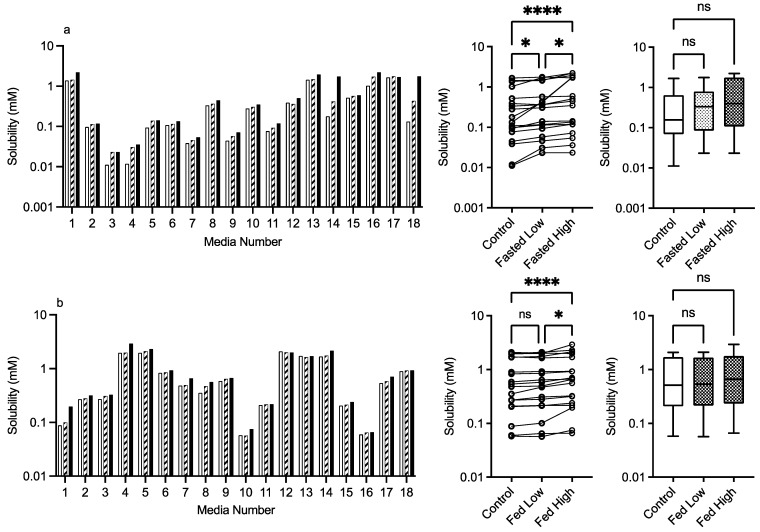
PVP K29/32. Top plots are for fasted media, bottom plots are for fed media. Plots from left to right: Bar graph shows individual media fenofibrate solubility measurements, 
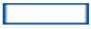
 control (no excipient), 
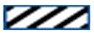
 low excipient concentration, 
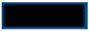
 high excipient concentration. Matched Friedman non-parametric comparison of groups (control, low excipient and high excipient concentration). Non-matched Kruskal–Wallis comparison of groups, 
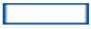
 control no excipient, 
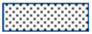
 low excipient concentration, 
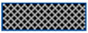
 high excipient concentration. Box and whisker plots—see legend in Figure 1. (**a**) ns = no significant difference, *p* > 0.05; * *p* = 0.0138; **** *p* < 0.0001. (**b**) ns = no significant difference, *p* > 0.05; * *p* = 0.0138; **** *p* < 0.0001.

#### 3.2.4. HPMC E3 and E50

The results for HPMC E3 and E50 are presented in Figure 6 and Figure 7, respectively. For this excipient, there is a very obvious solubility impact in the bar charts that is statistically significant for both comparative analyses, with the exception of the Kruskal–Wallis comparison of HPMC E3 in the fasted state. For HPMC E3, in the fasted and fed states for several of the media (e.g., 4 and others), the low excipient concentration increases solubility, which then decreases in the high excipient concentration. This concentration-dependent increase then decrease is not as prevalent for HPMC E50, but in both states, it is visually detectable in the Friedman comparison (Figure 6a,b). The Kruskal–Wallis figures for HPMC E3 and E50 indicate that in the high excipient concentration system, the entire solubility range is reduced significantly in three out of the four data sets. For this excipient in the standardised effect profile (Table 2), HPMC E3 in the low concentration is identical to the control, but all other systems have a changed profile. pH becomes non-significant as a factor, and for HPMC, E3 in the fasted state bile salt becomes significant, as does lecithin in the fed state. Overall, the results indicate that HPMC has a concentration- and molecular-weight-dependent impact on fenofibrate equilibrium solubility, which varies with media composition. 

**Figure 6 pharmaceutics-15-02484-f006:**
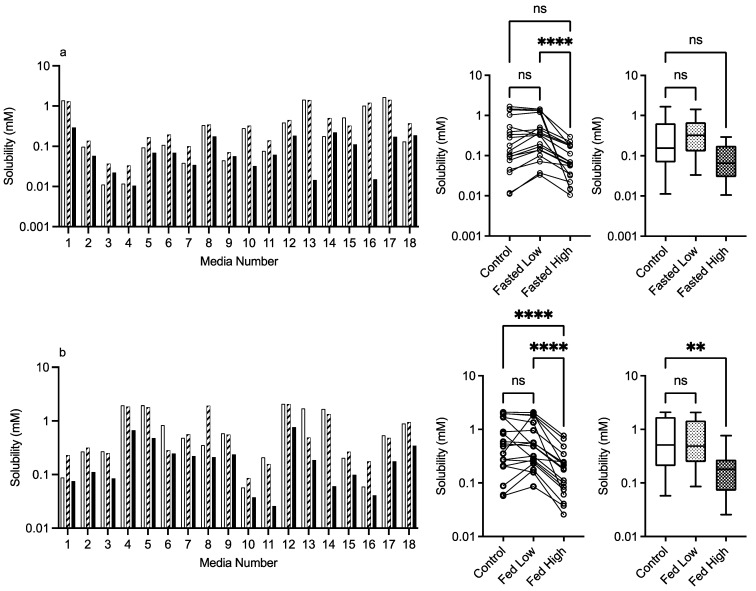
HPMC E3. Top plots are for fasted media, bottom plots are for fed media. Plots from left to right: Bar graph shows individual media fenofibrate solubility measurements, 
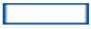
 control (no excipient), 
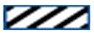
 low excipient concentration, 
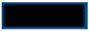
 high excipient concentration. Matched Friedman non-parametric comparison of groups (control, low excipient and high excipient concentration). Non-matched Kruskal–Wallis comparison of groups, 
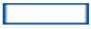
 control no excipient, 
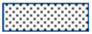
 low excipient concentration, 
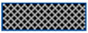
 high excipient concentration. Box and whisker plots—see legend in Figure 1. (**a**) ns = no significant difference, *p* > 0.05; **** *p* < 0.0001. (**b**) ns = no significant difference, *p* > 0.05; ** *p* = 0.0077; **** *p* < 0.0001.

**Figure 7 pharmaceutics-15-02484-f007:**
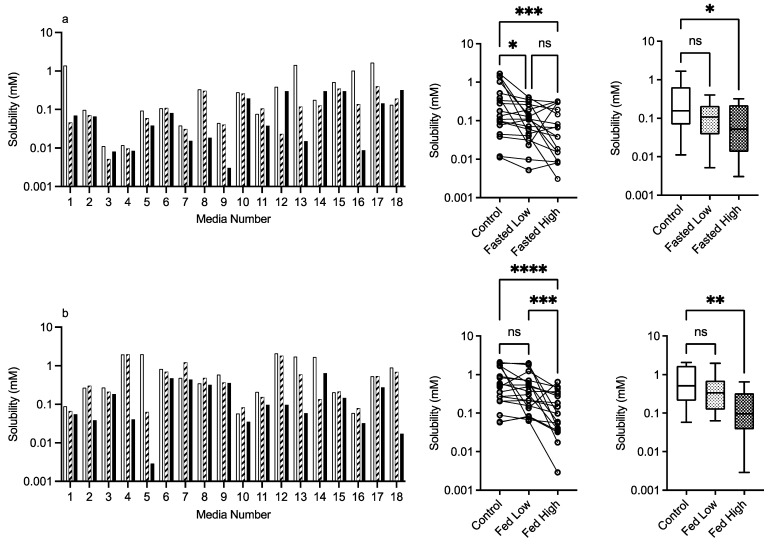
HPMC E50. Top plots are for fasted media, bottom plots are for fed media. Plots from left to right: Bar graph shows individual media fenofibrate solubility measurements, 
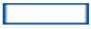
 control (no excipient), 
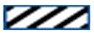
 low excipient concentration, 
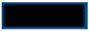
 high excipient concentration. Matched Friedman non-parametric comparison of groups (control, low excipient and high excipient concentration). Non-matched Kruskal–Wallis comparison of groups, 
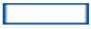
 control no excipient, 
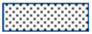
 low excipient concentration, 
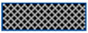
 high excipient concentration. Box and whisker plots—see legend in Figure 1. (**a**) Friedman comparison: ns = no significant difference, *p* > 0.05; * *p* < 0.0373; *** *p* < 0.0002; Kruskal–Wallis comparison, * *p* < 0.0263. (**b**) ns = no significant difference, *p* > 0.05; *** *p* = 0.0002; **** *p* < 0.0001; ** *p* = 0.0020.

#### 3.2.5. Chitosan

The results for chitosan are presented in Figure 8, and in a similar fashion to HPMC, the bar chart visually demonstrates a solubility impact in the media systems with all of the statistical comparisons registering at least one significant difference between the control and excipient. In the fasted system, there is a mix of behaviours especially in the low chitosan concentration, in which some systems increase while others decrease fenofibrate solubility causing the overall impact to be a reduction in the solubility range. The high chitosan concentration in the fasted state shows a significant reduction in solubility. This is also evident in the fed sate, with the solubility reduction related to the concentration of the chitosan present. This is reflected in the heat maps (Figure 9 and Figure 10), which show that the majority of the signals are negative.

**Figure 8 pharmaceutics-15-02484-f008:**
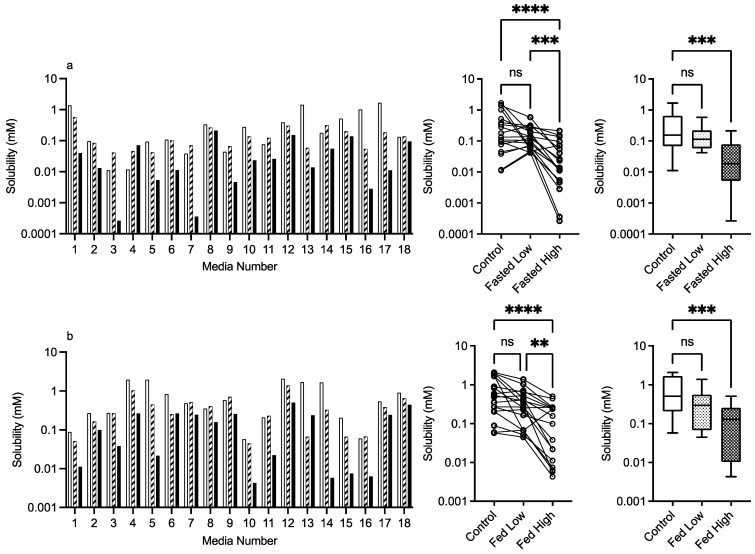
Chitosan. Top plots are for fasted media, bottom plots are for fed media. Plots from left to right: Bar graph shows individual media fenofibrate solubility measurements, 
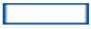
 control (no excipient), 
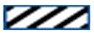
 low excipient concentration, 
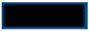
 high excipient concentration. Matched Friedman non-parametric comparison of groups (control, low excipient and high excipient concentration). Non-matched Kruskal–Wallis comparison of groups, 
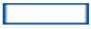
 control no excipient, 
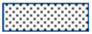
 low excipient concentration, 
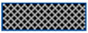
 high excipient concentration. Box and whisker plots—see legend in Figure 1. (**a**) Friedman comparison: ns = no significant difference, *p* > 0.05; *** *p* < 0.0007; **** *p* < 0.0001; Kruskal–Wallis comparison, *** *p* < 0.0004. (**b**) ns = no significant difference, *p* > 0.05; ** *p* = 0.0042; **** *p* < 0.0001; Kruskal–Wallis comparison, *** *p* < 0.0004.

**Figure 9 pharmaceutics-15-02484-f009:**
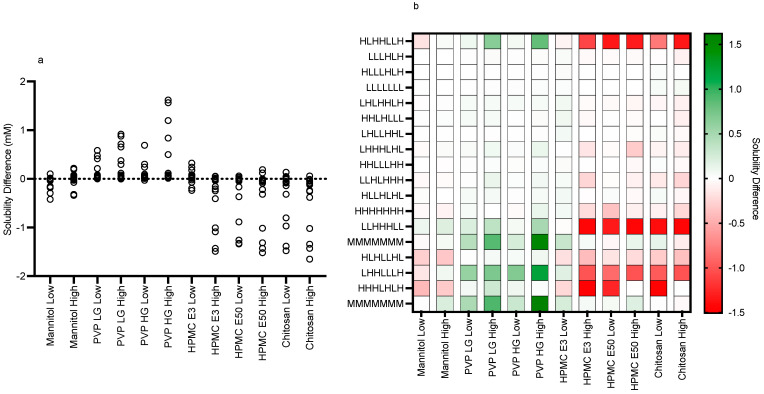
Fasted solubility difference and heat map. (**a**) Individual Excipient Measurements. (**b**) Heat Map of Solubility Differences vs Excipient and Media Composition. H, High media component level/concentration; M, Mid media component level/concentration; L, Low media component level/concentration (see Table 1). Media component order (from the left): pH, free fatty acid (sodium oleate), bile salt (sodium taurocholate), phospholipid (lecithin), monoglyceride (glyceryl monooleate), cholesterol, bile salt/phospholipid ratio.

**Figure 10 pharmaceutics-15-02484-f010:**
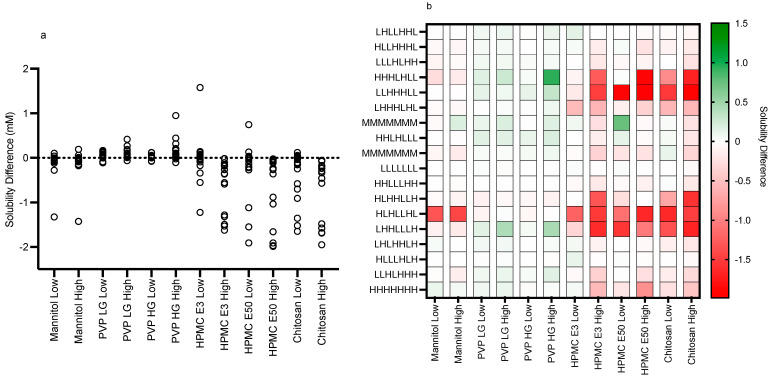
Fed solubility difference and heat map. (**a**) Individual Excipient Measurements. (**b**) Heat Map of Solubility Differences vs Excipient and Media Composition. H, High media component level/concentration; M, Mid media component level/concentration; L, Low media component level/concentration (see Table 1). Media component order (from the left): pH, free fatty acid (sodium oleate), bile salt (sodium taurocholate), phospholipid (lecithin), monoglyceride (glyceryl monooleate), cholesterol, bile salt/phospholipid ratio.

## 4. Discussion

### 4.1. Control Excipient-Free Solubility Measurements

The comparison indicates that there is a statistically significant difference between the fasted solubility data set from this study and that from a previously published fasted DoE study, but that there is no difference between the fed systems (Figure 1). The result for the fasted system is similar to previous comparisons [13] between a small experiment number dual-level DoE and the larger-experiment-number fasted versions, of which the statistical equivalence between the fasted and fed results in this study is not. However, the concentration levels applied in the different studies are not identical; in the larger-experimental-number fasted study [11], the phospholipid high level is higher (1.0 mM vs. 0.75 mM), free fatty acid is lower (10 mM vs. 15 mM), and cholesterol and monoglyceride are not present. There are also differences between the fed systems; in the larger-experimental-number study [12], bile salt is higher (24 mM vs. 15 mM), phospholipid is higher (4.8 mM vs. 3.75 mM), free fatty acid is higher (52 mM vs. 25 mM), monoglyceride is lower (6.5 mM vs. 9 mM) and cholesterol is absent. There is no difference in the pH range (5–7) between any of the systems. It is known that the solubility of neutral drugs in these systems is controlled by the total amphiphile concentration [11,12]. In addition, it is recognised that the application of a DoE approach to intestinal fluids, which links the high and low concentration levels of factors, creates media systems that are not equivalent and induce high levels of solubility variability [15]. 

Statistical analysis calculates that all of the data sets in this study had non-normal distributions (see Figure 1 for control, results not included in subsequent figures). This result has been reported previously in equilibrium solubility DoE studies [13,21] of simulated intestinal fluids. It has been attributed to the non-normal sample pattern induced by the DoE structure, resulting in non-normally distributed drug solubility values throughout the sample space. The small number of measurements in this study (n = 18 per state) could also be a statistical contributor, but even large DoE studies (66 or 92 samples) result in non-normal distributions (see Figure 1). Therefore, to compare the results within either a fasted or fed state, a non-parametric statistical comparison was applied. The experimental protocol using single media measurements introduces the potential for spurious statistical results, which necessitates a dual statistical analysis (Figure 3, Figure 4, Figure 5, Figure 6, Figure 7 and Figure 8). This issue can be resolved by repeated analysis of individual media systems, with current studies indicating that three replicates per system is sufficient [14,15], but this would increase the number of measurements in the study from 234 to 702 per state or 1,404 for the full protocol. This high experimental demand would require automation, or more realistically protocol modification, to target and maximise data gathering.

The detection of only two significant standardised effect values (Figure 2) is in marked contrast to the large scale DoE [11], in which five significant factors were identified for the fasted state, pH and buffer salts had a negative impact, and sodium oleate, lecithin and bile salt had a positive impact on solubility. In the fed state [12], five significant factors were also identified, bile salt had a negative effect, and sodium oleate, lecithin and monoglyceride had a positive impact. The result in this study is, however, comparable to a previous small-scale DoE study [21] that examined the fasted and fed states and only identified sodium oleate as a significant factor in both states. This is an inevitable outcome of reducing the number of experiments, which reduces the DoE’s statistical power and limits the assessment of an excipient’s impact on individual media factors.

The differences detected between the excipient-free values in this study and published studies is therefore consistent with the overall behaviour of simulated intestinal media examined using a reduced-experiment-number DoE approach. Despite these differences, since the system without excipients is acting as an internal control for this study, the impact of excipients on solubility can be directly analysed. However, based on the results in this study and other published studies surrounding system variability, the general applicability of excipient affects to all intestinal media systems must be examined cautiously. 

### 4.2. Effect of Excipients on Fenofibrate Equilibrium Solubility

#### 4.2.1. Mannitol

As far as we can determine, there are no previous published studies assessing the solubility impact of excipients in simulated intestinal media systems using a DoE approach. Therefore, the literature comparisons of results are limited to standard simulated intestinal media recipes. For mannitol, the detection of a significant change in the fed state using the Freidman analysis is potentially due to the experimental protocol with only a single measurement on each media system coupled with the non-parametric analysis (see Section 3.1.3 statistical considerations). All other results indicate that mannitol in the media systems has no impact on fenofibrate solubility. Mannitol as a neutral non-reducing sugar was not expected to affect solubility and has only been shown to influence drug solubility when the concentration is increased [22]. This result indicates that mannitol as an excipient will have a neutral effect on solubility in all intestinal media. 

#### 4.2.2. PVP K12 and K29/32

Both PVP K12 and K29/32 significantly increase fenofibrate equilibrium solubility (Figure 4, Figure 5, Figure 9 and Figure 10) in both the fasted and fed states in a concentration- and grade (molecular weight)-dependent manner. In addition, the solubility increase is media-composition dependent, larger in the fasted than fed state, and in the majority of cases, the solubility differences were near to zero. However, some solubility differences were positive in media with high levels of sodium oleate and high or low pH levels. Moreover, in the fed state the solubility difference of one media is increased by approximately 1 mM, with this media containing high levels of bile salt, sodium oleate and pH. These are media factors already known to influence fenofibrate solubility [11,12]. Therefore, PVP may simply be acting to synergise the media’s solubilisation potential. PVP is a water-soluble polymer that tends to increase drug solubility [23] and dissolution through a surface-tension-lowering effect [22]. A concentration-dependent influence of an excipient on solubility is to be expected [24] in simple solution systems and complex media systems in which only the excipient concentration is varied [25]. However, PVP K25 (MWt 30,000) had no effect on fenofibrate supersaturation in either FaSSIF or FeSSIF [26], and if supersaturation is linked to equilibrium solubility [16], this indicates no major effect due to the polymer. It should be noted that precipitation is a different property to solubility. PVP has been reported to interact with bile salt (sodium taurocholate) in both FaSSIF and FeSSIF to form polymer–bile salt aggregates in a concentration-dependent manner that was also influenced by media pH and ionic strength [27]. The authors comment that this could also impact solubility, and an interesting observation based on the Friedman comparison and heat maps is that the solubility increases media dependent nature. This is likely to be related to a specific interaction of PVP with the various media components and combination ratios, which is relatable to the previous finding [27]. It is also worth noting that the interaction between PVP K12 or K29/32 and the media increases solubility and no systems (based on the heat maps) present a noticeable solubility reduction. This may indicate that PVP can selectively and non-detrimentally support solubility across the intestinal media space. 

#### 4.2.3. HPMC E3 and E50

HPMC E3 and E50 exhibit an obvious solubility impact (Figure 6, Figure 7, Figure 9 and Figure 10) that is statistically significant in both comparative analyses, with the exception of HPMC E3 in the fasted media Kruskal–Wallis test, despite the obvious visual changes between the control and high HPMC E3 excipient concentration. The significant Kruskal–Wallis results between the control and high excipient concentration for the remaining HPMC results indicate that the polymer is reducing the entire solubility profile, especially at the high concentration. An interesting finding is the duality of HMPC E3′s solubility impact with the lowest concentration in the fasted state increasing but also decreasing solubility. HPMC is a water-soluble, hydrophilic, non-ionic cellulose ether available in different grades and viscosities [28] and has been shown to increase drug solubility and dissolution rate through the enhancement of drug wettability and decreasing surface tension. HPMC E5 and E50 have no impact on the supersaturation potential of fenofibrate in FaSSIF or FeSSIF [26], and HMPC K4M has also been demonstrated to interact with bile salts in FaSSIF or FeSSIF to form aggregates in a concentration-dependent manner that is also influenced by media pH and ionic strength [27]. As mentioned above, for PVP, this could impact solubility, and the interesting observation based on the Friedman comparison and heat maps is the effect’s media-dependent nature. This is likely to be related to a specific interaction of HPMC with the various media components and combination ratios [27]. Based on the heat maps, a common feature of a negative solubility impact is a high bile salt level coupled with a low cholesterol level. The standardised effect analysis for HPMC, in contrast to mannitol and PVP, is different to the control for HPMC E3 in the high concentration and E50 at both concentrations. In these latter cases, pH is no longer significant as a factor, and bile salt becomes significant for HPMC E3 in the high concentration in both fasted and fed states. This observation is consistent with the previous literature [27] on media aggregate formation and indicates that cholesterol, which is not present in standard FaSSIF and FeSSIF recipes, might stabilise bile salts against this effect. Due to the statistical limitations of the small-scale DoE design, see Section 4.1 above, this interpretation should be treated with caution but suggests that media composition is important in these studies. HPMC behaves differently to PVP, increasing, but mainly decreasing, solubility, indicating that “Janus”-type behaviour is possible, which further increases the difficultly of assessing its impact on intestinal solubility. Since polysaccharide-based excipients with variable grades and physicochemical properties are common in oral formulations [20], assessing their potential for media-based solubility interactions is warranted.

#### 4.2.4. Chitosan

Chitosan is a naturally occurring mucopolysaccharide of high molecular weight obtained by the alkaline deacetylation of chitin. Its weak solubility in water increases under acidic conditions due to the protonation of the amino groups with a pKa between 6.3 and 7.2 depending on the degree of deacetylation [29] and will therefore have a degree of ionisation at the media pH levels (5–7) used in this study. Since fenofibrate is neutral, it will not undergo a charge interaction with chitosan; however, charged media components are likely to interact. Chitosan exhibits an obvious concentration-dependent solubility impact (Figure 8, Figure 9 and Figure 10) that is statistically significant in all comparative analyses in the fasted and fed states. The heat maps indicate that the major impact of chitosan is in media, with a high bile salt level indicating an ionic interaction between the two materials, a feature that is also reflected in the changed standardised effect profile. This is consistent with a literature study that found acyclovir absorption was reduced due to the interaction of chitosan with luminal bile salts [30]. Additionally, the decreased absorption of water-insoluble drugs (indomethacin and griseofulvin) by chitosan [31] was attributed to the interaction of the cationic amino group of chitosan with the anionic group of fatty acids and bile salts. Therefore, chitosan as an excipient in oral formulations is likely to impact the bioavailability of poorly soluble drugs, especially those that rely on anionic amphiphilic media components for solubilisation. It should also be noted that chitosan has been investigated as a potential permeability enhancer [32], a feature that could offset the solubility reduction measured in this study. 

## 5. Conclusions

This is the first study to assess the feasibility of applying a reduced-experimental-number DoE protocol covering both fasted and fed simulated intestinal media to provide data on the impact of excipients on the equilibrium solubility of fenofibrate, a poorly soluble drug. Mannitol had no solubility impact in any of the DoE media. PVP significantly increased solubility in a media-, grade- and concentration-dependent manner, with the biggest change in fasted media. HPMC and chitosan significantly reduced solubility in both fasted and fed states in a media-, grade- and concentration-dependent manner. The results indicate that the impact of excipients on fenofibrate solubility is a complex interplay of media composition in combination with their physicochemical properties and concentration. The finding of media-composition-dependent behaviour indicates that excipient effects will be variable in vivo and complex to asses in vitro based on the inherent intra- and interindividual variability present in intestinal fluid [33]. The results also indicate that the combination of a small-scale DoE coupled with multiple excipients measured at two concentration levels is not optimal and future studies will require a balance of experimental load with data requirements, targeting either statistical or biorelevant information. Overall, the results indicate that assessing an excipient’s impact on a drug’s solubility in simulated intestinal fluid is feasible and provide interesting data that may guide formulation development.

## Figures and Tables

**Table 1 pharmaceutics-15-02484-t001:** Fasted and fed media components and concentration levels.

Component	MWt (g/mol)	Substance	Fasted State		Fed State	
			**Lower**	**Upper**	**Lower**	**Upper**
Bile salt	515.70	Sodium taurocholate	1.5 mM	5.9 mM	3.6 mM	15 mM
Lecithin	750.00	Phosphatidylcholine	0.2 mM	0.75 mM	0.5 mM	3.75 mM
Fatty acid	304.44	Sodium oleate	0.5 mM	15 mM	0.8 mM	25 mM
Mono-glyceride	358.57	Glyceryl monooleate	0.1 mM	2.8 mM	1 mM	9 mM
Cholesterol	386.65	Cholesterol	0.1 mM	0.26 mM	0.13 mM	1 mM
pH	Sodium hydroxide/hydrochloric acid qs		5	7	5	7
BS:PL ratio			7.5	7.9	7.2	4

**Table 2 pharmaceutics-15-02484-t002:** Standardised effect value results for excipient-containing media systems.

	Excipient/Concentration											
	Mannitol		PVP LG		PVP K29/32		HPMC E3		HPMC E50		Chitosan	
Media	Low	High	Low	High	Low	High	Low	High	Low	High	Low	High
Factor	Fasted											
pH	−S	−S	−S	−S	−S	−S	−S	NS	NS	NS	NS	NS
Na Oleate	+S	+S	+S	+S	+S	+S	+S	+S	+S	+S	+S	NS
Bile Salt	NS	NS	NS	NS	NS	NS	NS	+S	NS	NS	NS	NS
Lecithin	NS	NS	NS	NS	NS	NS	NS	NS	NS	NS	NS	+S
MG	NS	NS	NS	NS	NS	NS	NS	NS	NS	NS	NS	NS
Cholesterol	NS	NS	NS	NS	NS	NS	NS	NS	NS	NS	NS	NS
BS/PL ratio	NS	NS	NS	NS	NS	NS	NS	NS	NS	NS	NS	NS
	Fed											
pH	NS	NS	NS	NS	NS	NS	NS	NS	NS	NS	NS	−S
Na Oleate	NS	NS	NS	NS	NS	NS	NS	NS	NS	NS	NS	NS
Bile Salt	NS	NS	NS	NS	NS	NS	NS	NS	NS	NS	NS	NS
Lecithin	NS	NS	NS	NS	NS	NS	NS	+S	NS	NS	NS	+S
MG	NS	NS	NS	NS	NS	NS	NS	NS	NS	NS	NS	NS
Cholesterol	NS	NS	NS	NS	NS	NS	NS	NS	NS	NS	NS	NS
BS/PL ratio	NS	NS	NS	NS	NS	NS	NS	NS	NS	NS	NS	NS

S, factor significant in excipient system; NS, factor not significant in excipient system; −/+, direction of significant effect.

## Data Availability

All data presented this publication are openly available from the University of Strathclyde KnowledgeBase at https://doi.org/10.15129/253deccb-5e58-42f2-9791-24f79c71beb3.

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
