# Peer review of "Excipient Impact on Fenofibrate Equilibrium Solubility in Fasted and Fed Simulated Intestinal Fluids Assessed Using a Design of Experiment Protocol"

_pharmaceutics, 2023, doi:10.3390/pharmaceutics15102484_

Round 1
Reviewer 1 Report
Ms.Nr: pharmaceutics-2532194
B. E. Ainousah et. al: “The impact of excipients on fenofibrate equilibrium solubility in fasted and fed simulated intestinal fluids assessed using a Design of Experiment protocol”
Authors present a newer piece of their research on influence of biomimetic media components and excipients of oral formulations on solubility of BCS II drugs using DoE. This time fenofibrate (PF) serves as model compound. The contribution of authors to this scientific area is remarkable, what reflects well the 11/37 (30%) self-citations in this manuscript.
The aim of present work is to investigate the effects of excipients on the equilibrium solubility of fenofibrate in a 36-experiment dual range DoE with seven gastrointestinal media components (pH, sodium oleate, bile salt (BS), lecithin (PL), monoglyceride, cholesterol and BS:PL ratio) covering both fasted and fed states in a single experiment with four factor levels. The goal was to reduce the solubility sample number using statistical design of experiments.
The topic is important. To gain knowledge about the excipients effect on equilibrium solubility is raising great interest in both industry and academia. The work is comprehensive, the solubility measurement protocol and the statistical evaluation of results are correct. The text, despite the very complex experimental design, is clear and logically constructed, readable. The figures are insightful, particularly Fig. 9. and 10.
The conclusion clearly and correctly confesses, that the combination of a small size DoE to reduce experimental load coupled with multiple excipients measured at two concentration levels, is not optimal. So finally, the extensive work has led to negative outcome. Reviewer thinks that is not a failure, rather contributes to our overall knowledge on this field.
I suggest the acceptance for publication after minor revision.
Some critical remarks:
1. The pH selection is not enough clear. Fasted media has pH 6.8, fed 5.0. Why here at both states 5 and 7 were used? A detailed interpretation of this is needed.
2. Figure 1. shows significant difference in fasted state from the previous results, what explained in the text with different component concentrations used in the media. Why not the same concentrations were applied?
3. Figure 2. How can be interpreted that pH has effect on solubility of neutral PF in fasted but not in fed state in excipients free media? Which components can cause this?
4. The manuscript is enough lengthy. Putting Table 2-4 into Supplementary material would significantly reduce the length. However, a new table about the composition of media 1-18 would be very useful. Without this Fig. 9 and 10 y axis (HLHHLLH, etc) is not uderstandable.
5. Page 11. on Fig 5, the Fig. 6a and Fig. 6b marks are erroneous. On page 13 at line 345: Chitosan results are shown on Fig. 8, not Fig. 7.
Reviewer 2 Report
1) Title: 'Design of Experiment' can be written as 'design of experiment'. The usual abbreviation of 'DoE' may still be used.
2) Title: Remove the full stop after 'protocol'
3) I suggest that you add the structures of fenofibrate and mannitol, and the repeat units of PVP, chitosan, and HPMC.
4) Line 64. In future, if you used 2-level experiments, you could consider using a Plackett-Burman design as the initial design and then construct a 5-level D-optimal design. You will also perform only a few experiments in each design and be able to determine if there is curvature in the responses. (You should also be able to use the current results and add a few experiments to 'transform' the design to a different type. D-optimal designs typically include a factorial design component).
5) Line 92. I suggest to change 'neutral' to 'non-ionic'. There are always some extent of polarity which are independent of electrostatic forces.
6) Line 127. Do you mean 'technical' grade when you say 'practical grade'?
7) Please ensure/confirm that all units of measurement complies with the SI standard notations or IUPAC guidelines. For example 'ml' vs 'mL'.
8) Table 1 and 2 (check all tables) uses different notations for 'upper' and 'Upper', 'lower' and 'Lower' etc.
9) The font formatting in various subheadings is not used consistently. Compare, for example, headings 2.2 and 2.3.
10) Table 2. Maybe you don't need to add so many decimals. If you add decimals, keep the number consistent.
11) Line 164. You should number all subheadings.
12) Line 258. You should use a ':' and not a '.' to list the reasons.
13) Line 361. 'Excipient Free' should be 'Excipient-Free'.
The quality is high. Maybe a final screening should reveal more issues.
Reviewer 3 Report
The paper „The impact of excipients on fenofibrate equilibrium solubility in fasted and fed simulated intestinal fluids assessed using a Design of Experiment protocol“ is appropriate for Pharmaceutics in the terms of journal scope. However, some issues need to be addressed before publication.
1. The title should be reconsidered since there is Capitalized words in the middle of the title and a full stop at the end. „The excipients impact on fenofibrate equilibrium solubility in fasted and fed simulated intestinal fluids assessed using a experiment protocol design“ may be more appropriate.
2. The abstract needs to be rewritten since there are sentences that are not easy to follow. For instance „This study has applied a reduced experiment number design of experiment equilibrium solubility approach to investigate the impact of excipients on fenofibrate solubility in fasted and fed media.“ may be more clear as „This study has applied a reduced experiment number design of equilibrium solubility approach in order to investigate the excipients impact on fenofibrate solubility in fasted and fed media“. Also „Mannitol had no solubility impact but for the remaining excipients increased solubility whilst HPMC E3 and E50 along with chitosan reduced solubility in both fasted and fed states.“ is confusing. Remaining excipients increased solubility... which one if hydroxypropylmethylcellulose (HPMC, E3 and E50) and chitosan reduced solubility...
Further more, „The results indicate that the solubility impact of excipients is a complex interplay of media composition in combination with excipient physicochemical properties and concentration.“ may be better as „The results indicate that the impact of excipients on fenofibrate solubility is a complex interplay of media composition in combination with their physicochemical properties and concentration,“. Finally, the sentence „Appropriate in vitro studies solubility studies of drug, media composition and proposed excipients will provide interesting information to guide formulation development.“ is unclear „Appropriate in vitro studies solubility studies“ is this part refer to in vitro studies of solubility or in vitro studies and solubility studies... so confusing. There are no specific results given in the abstract. No statistically signifcance of solubility redcution and/or increment by the excipients is emphasized. The sentences are general.
Sodium oleate is not even mentioned in the abstract for example which has positive effect on solubility according to the results presented in this paper.
3. Line 48 „All with the aim of improving drug solubility and dissolution rates“. All what? Maybe „All abovementioned approaches are considered with the aim of improving drug solubility and dissolution rates.“
4. Lines from 83 to 120 need to be addressed very carefully since the introduction part is mixed with the aim of the study, the material and methods and even with a discussion part. The part where the authors explain the characteristics of various excipients in general belong to the introduction part. The introduction part should be finished with an aim of a study that is not defined in this manuscript. The part regarding the protocol belongs to the material and method while the explanation why different concentrations are applied and comparison with the data from other studies should be part of the discussion. This is a complete mishmash and needs to be systematized.
5. Line 211... the part with HPLC analysis requires far more details about the method validation: values for limit of detection, limit of quantification, recovery, linearity, sensitivity etc... for each compound, as well as their retention times in order to compare whether there were interferences of fenofibrate and each excipient...
6. Line 430-432 „This result indicates that mannitol as an excipient will have a neutral effect on solubility in all intestinal media systems a result which could possibly also apply to other simple sugars“. If you want your research to be considered valid, it is not scientific to have this kind of predictions without experiments. Please consider to delete the last part of the sentence that this results can be extrapolated to any neutral sugar.
7. Line 513 - This maybe a general property of all polymeric cationic excipients. Please delete this part. General assumptions without experimental evidences are not science and do not belong in this paper.
8. Conclusion needs to be rewritten. The authors have completely missed the point what conclusion stands for. Conclusion part should answer the aim of the study and to summarize the most important findings of THIS study and not to refer to other papers not to discuss other findings. The comparison with the results of other studies should be transferred to the discussion part and in this part the authors should be focused solely on the results obtained in their research. We need to conclude what this study has found, what is new and original? General sentences such as „Solubility impact can be related to the excipient’s chemical structure and grade and its potential interaction with simulated media components. The finding of media composition dependent behaviour indicates that excipient effects will be complex to asses in vitro and variable in vivo based on the inherent intra and inter individual variability present in intestinal fluid [36,37]. The utilisation of excipient and polymer mixes to increase solubility in specialized oral formulations [18] would add to the complexity“ do not belong to conclusion part or to this paper at all.
Minor correction of English should be considered. Some sentences are too long and hard to follow. Other sentences are not clear enough.
Round 2
Reviewer 2 Report
No comments
Minor editing is required
Reviewer 3 Report
The authors have made the required changes. The paper can be accepted as presented.